# Bacterial age distribution in soil – Generational gaps in adjacent hot and cold spots

**Benedict Borer** [id][¤a]*, **Dani Or**[¤b]

ETH Zurich, Zurich, Switzerland

¤a Current address: Massachusetts Institute of Technology, Cambridge, Massachusetts, United States of America
¤b Current address: Desert Research Institute, Reno, Nevada, United States of America
* bborer@mit.edu

**Data Availability Statement:** The code used for generating all data displayed in this manuscript is available in the supplementary material S1 Data.

**Funding:** Financial support for this work came from an Advanced Grant to D.O. by the European

## Abstract

Resource patchiness and aqueous phase fragmentation in soil may induce large differences local growth conditions at submillimeter scales. These are translated to vast differences in bacterial age from cells dividing every thirty minutes in close proximity to plant roots to very old cells experiencing negligible growth in adjacent nutrient poor patches. In this study, we link bacterial population demographics with localized soil and hydration conditions to predict emerging generation time distributions and estimate mean bacterial cell ages using mechanistic and heuristic models of bacterial life in soil. Results show heavy-tailed distributions of generation times that resemble a power law for certain conditions, suggesting that we may find bacterial cells of vastly different ages living side by side within small soil volumes. Our results imply that individual bacteria may exist concurrently with all of their ancestors, resulting in an archive of bacterial cells with traits that have been gained (and lost) throughout time–a feature unique to microbial life. This reservoir of bacterial strains and the potential for the reemergence of rare strains with specific functions may be critical for ecosystem stability and function.

## Author summary

The study addresses the simple question: "What is the average age of bacterial cells in soil". Limitations of current experimental methods in resolving g cell age distributions in soil samples, motivated the use of modeling approaches for linking soil physical and hydration conditions with localized bacterial cell demographics. In contrast with other life forms, bacterial cells may persist for long periods at subsistence state close to starvation. Kin cells in close proximity on the other hand may divide frequently due to high availability of nutrients, resulting in numerous daughter cells and an ever-growing generational gap. In human terms, we would be living concurrently with relatives from medieval times or even earlier. Our results suggest that although the majority of generation times of the order of hours and days, the age distribution shows a heavy tail with long generation times and very old cells persisting in proximal soil volumes. The variation in localized

Research Council (ERC-3200499-'SoilLife', https://erc.europa.eu/) and from the RTD SystemsX.ch project 'MicroscapesX' (http://www.systemsx.ch/). The funders had no role in study design, data collection and analysis, decision to publish, or preparation of the manuscript.

**Competing interests:** The authors have declared that no competing interests exist.

growth rates provides a living soil bacterial reservoir with the potential for the reemergence of rare strains that may contribute to ecosystem stability and function.

## Introduction

Notwithstanding the harsh and dynamic environmental conditions, soil microbial life thrives at all scales–with a single gram of fertile soil may contain up to $10^{10}$ prokaryotic cells [1]. Even with such high potential abundance, soil bacteria inhabit less than 1% of the available soil surface [2] and are largely associated with patchy and nutrient-rich soil volumes that may support densely populated hotspots of biological activity (the rhizosphere or the detritusphere within soil aggregates [3]). Within these hotspots, availability of nutrients results in high cell growth rates (similar to the growth of copiotrophic bacteria grown in laboratory settings [4]). In contrast, the remaining 99% of soil surfaces and volumes support act as biological "cold spots" with very little to no bacterial activity due to a lack of nutrients [5] and unfavorable hydration conditions [6]. Soil hydration status has been shown to be a key variable that governs multiple functions of microbial life such as community structure [7], horizontal gene transfer rate [8], cell dispersal [9–11] and nutrient diffusion (both aqueous and gaseous) [6]. From a nutrient flux perspective, too wet or too dry soil conditions are generally unfavorable for bacterial activity, due to either a lack of oxygen when saturated or limited aqueous nutrient diffusion in dry soil. Optimal nutrient and gaseous fluxes, and related high growth rates, are often supported at intermediate hydration levels [12]. Hydration conditions also dictate bacterial dispersal, where water saturated conditions often facilitate convection or motility through the pore network thus enabling relocation over large distances and introduction of bacterial cells to more favorable locations. In contrast, drier conditions and associated thin water films restrict bacterial cell motility and result in a fragmented habitat where pinned bacterial cells are restricted to diminishing diffusive nutrient fluxes and limited prospects for proliferation. In addition, the nature of different hotspots (e.g. degradation of recalcitrant carbon in the form of root detritus versus growth on root exudates in the rhizosphere) further broadens the growth rate distribution and community composition [13]. These contrasting conditions concerning bacterial growth rates may occur within small soil volumes and result in significant generation time disparity, with rapid cell proliferation coexisting next to nearly dormant bacterial cells that support their maintenance with limited prospects for growth and cell division. Even after episodic wetting events that reconnect bacterial habitats and permit temporary infusion of nutrients, subsequent internal drainage fragments the aqueous phase and growth rates drop following diminishing accessibility and availability of nutrients. This results in a scenario unique to the microbial world where the dynamics of a population is a function of localized conditions and the development stages of the population as a whole are preserved in bacterial cold spots–creating an accessible library of species-specific functional traits through time. In this study, we seek to understand the consequences of this common disparity in local bacterial growth rates in soil and impacts on cell lineage propagation and average bacterial cell ages in soil.

At present, there are no direct methods for inferring the ages of individual bacterial cells within a natural soil sample. We define cell age as the elapsed time since last cell division, and generation time as the cell age at division (also known as the interdivision time, doubling time or cell-cycle time [14]). Experimentally, cell age and related generation time distributions are deduced indirectly from average growth rates or microbial activity data. Multiple techniques have been used for measuring growth rates *in situ*, such as direct cell counts, radioactively labeled thymidine/leucine incorporation rates [15,16] or using observed and expected

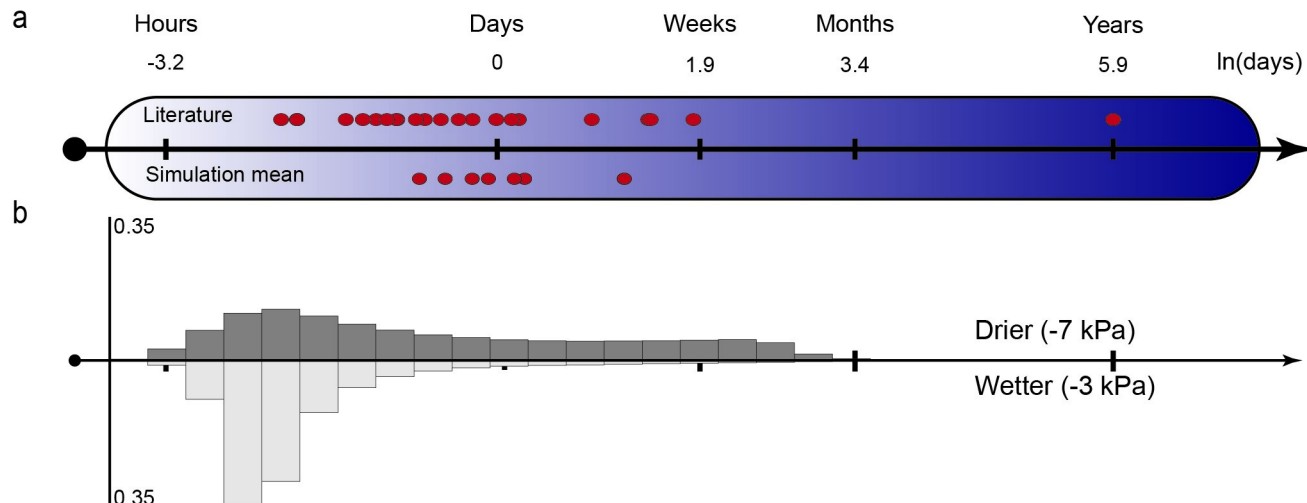

**Fig 1. Distribution of experimentally measured bacterial generation times in topsoil and the rhizosphere compared to computationally obtained, generation time distributions.** a) Comparison of literature values [4,20–26] and mean generations times in the simulations for different hydration conditions. Further information on the soil samples and quantification techniques of the literature values can be found in S1 Table. b) Probability distribution of generation times within a single simulation under different hydration conditions. Drier conditions (-7 kPa) promote a broader distribution with a more pronounced tail of bacterial generation times compared to intermediate hydration conditions (-3 kPa) where rapid growth is enabled due to optimal gaseous and aqueous diffusion rates.

mutation accumulation rates [17]. More recently, omics-based methods, such as quantitative stable isotope probing (qSIP) using $H_2O^{18}$, have emerged as a novel technique to measure growth rates in environmental samples [18] and have even enabled the quantification of growth rates at the taxon level [19]. Using these techniques, mean generation times of soil microbial communities have been determined for a wide range of conditions (Fig 1).

A critical drawback of the above-mentioned experimental techniques is that the estimated growth rates in soil are sample mean values that do not resolve the within sample age distribution [15]. Additionally, most methods suffer from poorly resolved detection limits that favor identification of rapid growth, thereby biasing the demographic picture in favor of younger cells. Theoretically, we expect bacterial life in patchy soil microenvironments to produce far broader age distributions than could be resolved by current measurement methods. While direct experimental evidence from soil is scarce, we can gain insights from analogous patchy environments such as bacterial age in the phyllosphere [27]. Studies have tracked the reproductive success (the total number of offspring produced by an individual) of isogenic bacterial cells by linking these to leaf patchy nutrient distribution and overall local carrying capacity [27]. These experiments have shown a broad distribution of reproductive success (and thus generation times) for individual cell lineages despite their isogenic characteristics, which supports the role of contrasting growth conditions in soil. We hypothesize that microscale spatial variations in soil hydration state or nutrient distributions would shape the age and generation time distribution of bacterial cells with broader distributions emerging in drier conditions due to habitat fragmentation and nutrient flux limitations. To test this hypothesis, we employ a previously published modeling framework that combines the salient features of soil aqueous phase configurations with individual-based bacterial cell growth and dispersal [28]. The simulations consider a simple scenario of a motile (by means of a chemotactically biased run-and-tumble flagellated motion), obligate aerobic bacterial species growing on a single carbon source and oxygen following Monod kinetics (see details in the Modeling section). The model tracks the life history of each bacterial cell within the simulation domain, and its lineage with

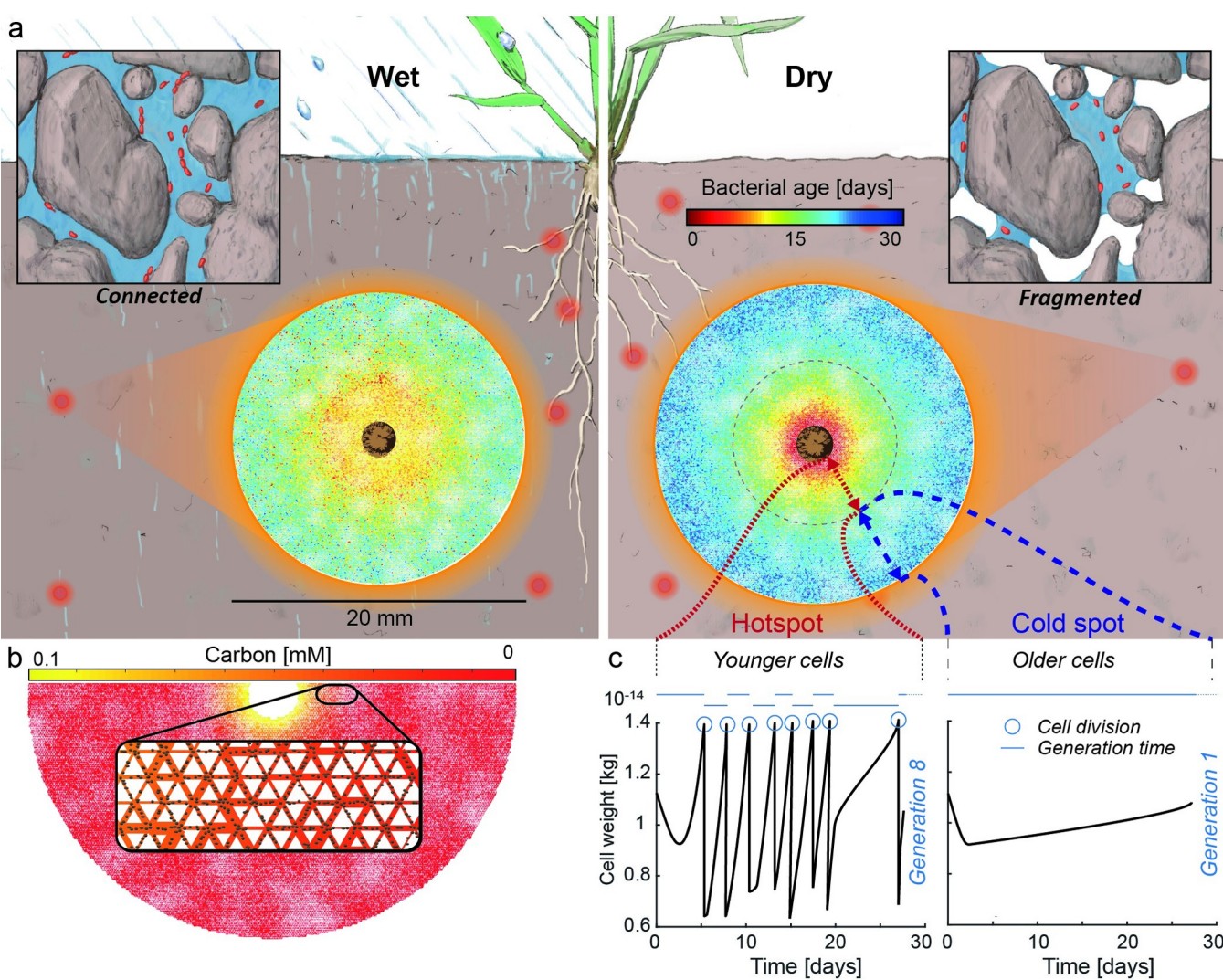

**Fig 2. Bacterial population demographics shaped by diffusion and dispersal limitations around soil hot spot as affected by hydration conditions.** a) Spatial visualization of simulated bacterial cell ages in the whole simulated domain of 20 mm diameter for -3 kPa and -7 kPa representing wet and dry conditions, respectively. Diffusion and dispersal restrictions in dry conditions expose individual lineages to harsh conditions resulting in older cells growing close to their maintenance rate. b) The model imitates a soil bacterial hotspot with a central carbon source and peripheral oxygen. Bacterial cells are represented as individual agents inhabiting an angular pore network with varying pore sizes. Diffusion of aqueous substrates and cell dispersal is dictated by thin water films within the angular pores, resulting in a patchy nutrient landscape and localized growth conditions. c) The spatially patchy resource landscape gives rise to bacterial "hotspots" (nutrient rich and accessible by a well-connected aqueous phase) and "cold spots" reflecting nutrient limitations and fragmentation. Cell lineages inhabiting hotspots proliferate and attain high reproductive success whereas (kin) isogenic lineages in cold spots persist with minimal prospects for growth and dispersal.

reference to the localized nutrient conditions experienced (Fig 2). This enables attribution of the (simulated) life history of each cell to local conditions and to its age and generation time, thus providing estimates of community age and generation time distributions by integrating across the bacterial population. The soil-like landscape and the aqueous phase distribution within it vary with hydration state, giving rise to limited diffusive fluxes imposed by thin water films that support survival of bacterial cells near their maintenance rate whilst neighboring cells close to nutrient sources may proliferate at near maximum growth rates (Fig 2). While the numerical and analytical models are instrumental in providing insights into drivers and modifiers of bacterial population age and generation time distributions, their short time scales

and numerous simplifying assumptions regarding bacterial traits, diversity and gene flow preclude their use for making inferences regarding potential evolutionary processes or shifts in population diversity, and we thus treat such potential generalizations as mere speculations. In addition to mechanistic modeling of emerging bacterial generation time distributions in a soil-like system, we capitalize on a heuristic analytical formulation for linking local heterogeneity in bacterial growth conditions (nutrients and hydration) to the resulting bacterial age distribution.

## Results

Fig 3A depicts the population size at the end of the simulation as a function of hydration state when simulated using the IndiMeSH model (details of the simulation conditions and parameters are found in the methods section and S2 Table). In this study, we use the soil water matric potential that represents the energy state of soil water to represent prevailing hydration conditions (more negative values imply drier soil). The matric potential is not only linked to the amount of water in soil pores, it also controls its organization in soil pores and film thicknesses that support diffusive fluxes of nutrients. For saturated conditions (matric potential of 0 kPa and -1 kPa), growth of the obligate aerobic bacterial community is restricted by low oxygen diffusion through the water saturated pores limiting growth of the obligate aerobic bacteria. Intermediate hydration conditions (-2 kPa and -3 kPa) create an optimal balance of aqueous and gaseous nutrient diffusion that enables rapid growth. As drier conditions set in, thin water films limit aqueous nutrient diffusion thereby reducing bacterial community size. Fig 3B shows the empirical cumulative distribution function of the cell generation times during the simulation. On average, the shortest generation times were realized at -2 kPa to -3 kPa (optimal balance between gaseous and aqueous nutrient diffusion) and diverged towards longer mean generation times for wetter or drier conditions (Fig 3B and S3 Table).

For simplicity, we focus in this study on isogenic bacterial populations and their spatially distributed responses to conditions similar to a soil microbial hotspot and its surroundings where high nutrient concentrations support rapid cell proliferation near the nutrient source. The majority of bacterial cell generation times realized in a simulated hotspot are in the order of hours to days for most hydration conditions (prescribed maximum growth rate results in a shortest generation time at optimal conditions of 0.5 h). However, the generation time distributions are characterized by persistence of very long generation times (a heavy tail), as shown in Fig 3B. Interestingly, we find that under wet conditions (Fig 3C), the tail of the distribution is truncated (i.e. fewer long generation times) due to bacterial motility through the water-saturated pore spaces that enable them to relocate towards more favorable conditions [29]. On the other hand, low oxygen diffusion rates restricted by saturated pore spaces result in anoxic conditions that shift the distribution towards longer generation times as shown be the lower frequency of very short generation times in Fig 3C. Intermediate conditions (-3 kPa, Fig 3D) support rapid proliferation of the bacterial community due to optimal nutrient fluxes. Under drier conditions (Fig 3E), cell motility is greatly reduced and the dispersal range is restricted [11] with concurrent reduction in nutrient diffusion rates that result in a wide distribution of generation times. S1 Fig shows the original and extrapolated simulation data in relation to multiple heavy-tailed distributions (power law distribution, exponentially truncated power law distribution, exponential distribution, gamma distribution and lognormal distribution) for all hydration conditions. The distributions of generation times in Figs 3C, 3D, and S1 were extrapolated by calculating the time required to complete a life cycle (generate the necessary biomass to cell division) based on observed individual growth rates at the end of the simulations. More precisely, we calculated the time required to assimilate the difference from current

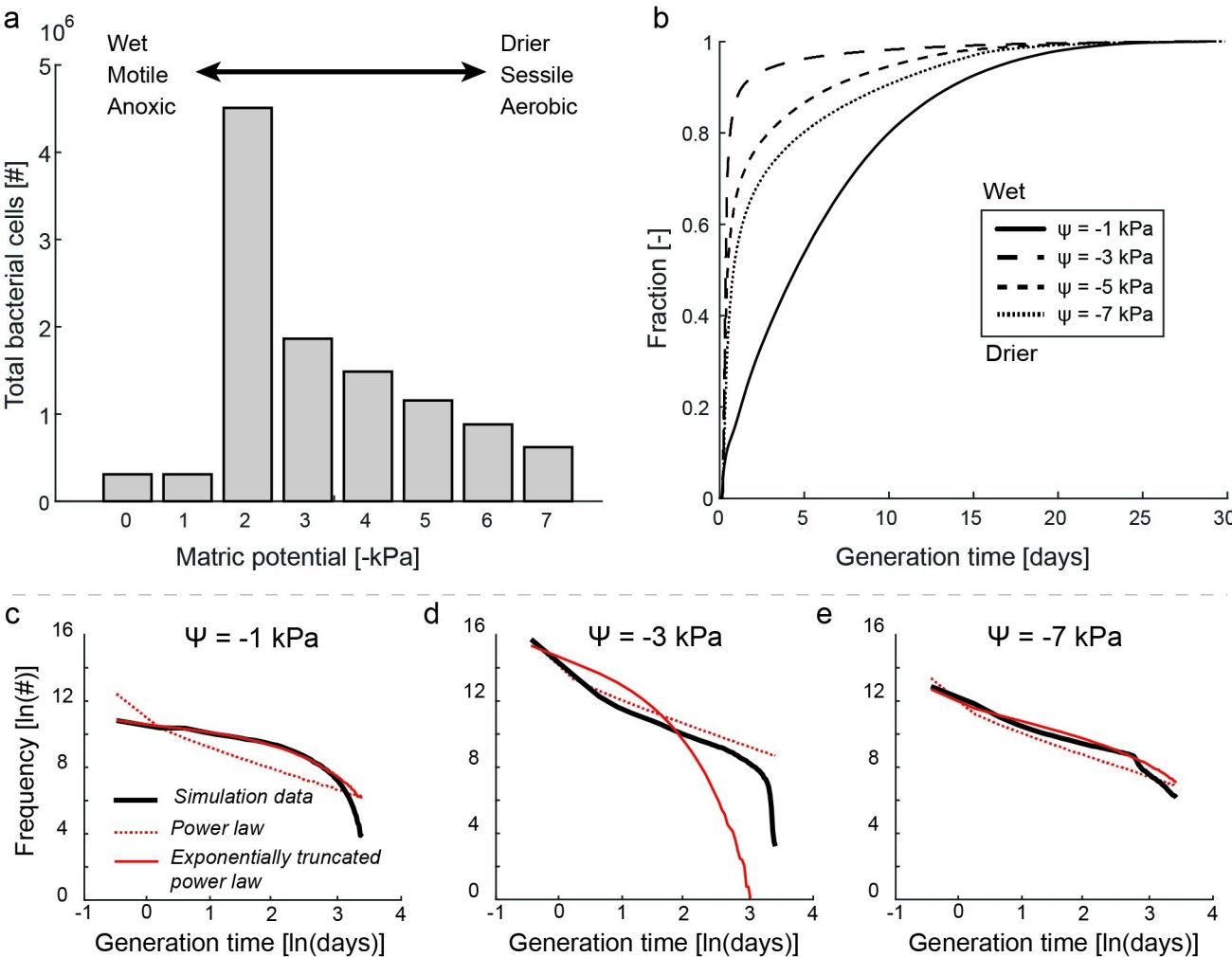

**Fig 3. Variations in simulated bacterial population sizes and generation time distributions as a function of hydration conditions.** a) Final simulated population size depending on hydration conditions where optimal growth conditions occur at intermediate hydration conditions that support sufficient diffusion of both gaseous and aqueous nutrients. b) Empirical cumulative distributions of observed generation times during the simulated time. On average, the shortest generation times were obtained at -2 and -3 kPa due to optimal growth conditions. The cumulative density function curves for generation times at 0 kPa and -1 kPa are congruent. Log-log visualization of the extrapolated simulation data with fitted power law and exponentially truncated power law distributions for the wet (c), intermediate (d) and drier (e) scenarios. In saturated conditions (0 and -1 kPa), bacterial motility enables cells to relocate towards more favorable conditions, resulting in a more truncated tail of the distribution that is better described by the exponentially truncated power law distribution. In addition, emerging anoxic conditions restrict rapid proliferation of the simulated obligate aerobes, skewing the distribution towards a longer average generation time. Intermediate conditions (-2 kPa and -3 kPa) still enable relocation of bacterial cells whilst permitting optimal aqueous and gaseous diffusion, resulting in rapid proliferation of the population. In drier conditions (-4 kPa and higher), bacterial motility is restricted and habitat fragmentation results in spatially isolated subpopulations growing at vastly different growth rates giving rise to the wide distribution in generation times and a more pronounced tail of the distribution that is captured by a power law.

cell biomass to the biomass at division for each cell using localized growth rate of each cell at the end of the simulation. These estimated times are then added to the current cell age for predicting the most likely generation time under present and local conditions. We find that the main influence of hydration conditions lies in promoting the emergence of a more pronounced tail of the distribution (following diminishing nutrient conditions in drier conditions). The choice of obligate aerobe as model bacterium implies sensitivity to saturated conditions with oxygen diffusion limitations irrespective of location and the resulting age distribution under these conditions must be interpreted with caution.

A heavy-tailed generation time distribution suggests a broad range of reproductive success where a few lineages dominate the total bacterial biomass due to rapid proliferation whereas lineages with long generation times contribute little to the total biomass of a population. In this context, a cell lineage is defined as the inoculated cell at the simulation initialization and all of its progeny with the reproductive success being the maximum generation achieved by a single lineage during the simulated time. Lineages with high reproductive success are those very few inoculated cells and their progeny that contribute disproportionally to the final population biomass, and are further defined as the dominant fraction. In contrast, a large proportion of the inoculated cells are incapable of proliferating and only contribute a minute fraction to the final biomass (rare fraction). To classify lineages into dominant and rare, we use a minimum cross-entropy algorithm originally developed for image thresholding [30]. Fig 4A shows the fraction of the initial inoculum classified as either dominant or rare lineage. Interestingly, motility under wet conditions plays an important role as it enables individual cells to relocate towards favorable conditions, thereby equilibrating reproductive success between lineages rendering most of lineages as dominant. Under dry conditions, the combination of diffusion-limiting thin water films with pinning forces that limit cell dispersal ranges [6,11] result in proliferation of only a few lineages that are close to the carbon source. More generally, conditions that support motion and migration equalize the contribution of lineages whereas restrictive conditions that limit dispersal and resources patchiness favor a few rare lineages. For all hydration conditions, the summed contribution of the rare lineages is less than 3% of the final population biomass (Fig 4A). This stark difference in reproductive success between the rare and dominant lineages results from equally prominent differences in their mean age distribution (Fig 4B) and mean generation times (Fig 4C) of these lineages. The mean cell age of individual lineages diverges towards older cells for rare lineages and younger cells for dominant lineages with increasing matric potential (drier conditions) (Fig 4B). The observed mean young cell age for rare lineages primarily stem from lineages that divided very late in the simulation due to growth close to the maintenance rate. Thus, these results suggest a bimodal distribution of bacterial age distributions in soil for bacterial cells growing close to a carbon source (hotspot) and within the bulk soil (cold spots) especially for drier conditions (S2 Fig). Similarly, there is a relationship between the average generation time and the relative abundance of individual lineages (Fig 4C). Overall, rare lineages have a longer average generation time compared to dominant lineages, which is expected theoretically when considering the different reproductive success of the two groups.

Finally, we derive a mathematical description of how the heavy-tailed distribution of cell ages in a soil community may arise from a diversity in growth rates of individual populations. For a single isogenic population growing under steady state conditions, the cell age distribution is expected to follow a modified exponential decay distribution [31]:

$$u(a) = 2 \cdot k \cdot e^{-ka} \cdot \int_a^\infty \omega(a) \cdot da \qquad (1)$$

where u(a) is the probability of a cell having age a, k is the growth rate and $\omega(a)$ is the probability distribution of generation times with the integral $\int_a^\infty \omega(a) da$ the survival function. This derivation assumes that steady state conditions have prevailed for at least as the oldest observed cell age within the population [31], a symmetric division of the bacterial cells (i.e. equal daughter cells) [32] and that death is negligible in the bacterial population [31]. These assumptions delineate an important upper boundary for the subsequent interpretation of the emerging cell age distributions.

Frequently, the observed generation time distribution for a population grown in homogeneous environments is best described by a Gamma distribution [32–34] with shape parameter

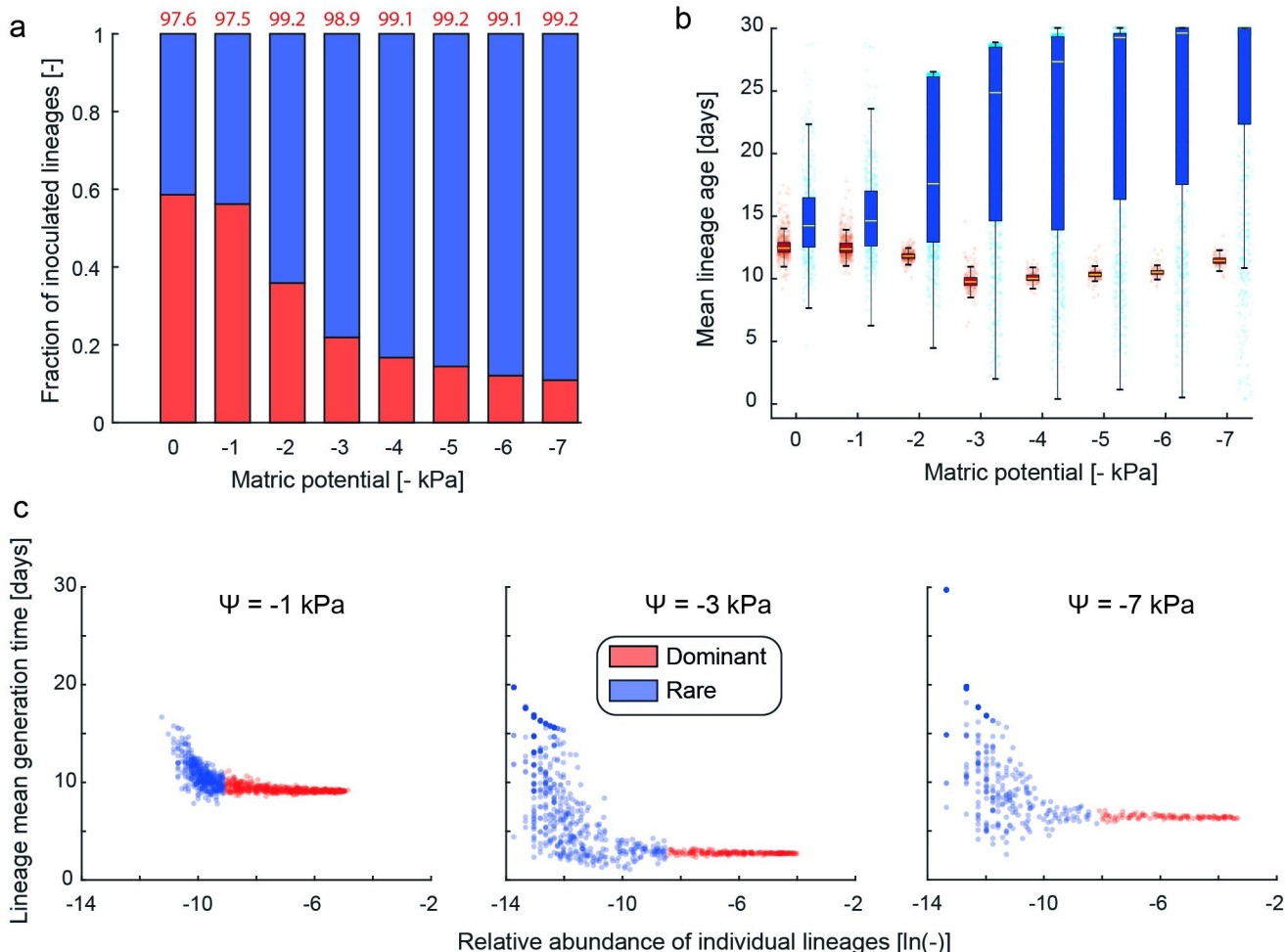

**Fig 4. Characteristics of dominant (highest reproductive success) and rare lineages vary with hydration conditions.** a) Percentage of lineages from the inoculum classified as dominant and rare. With a decline in bacterial motility towards drier conditions, most inoculated lineages cannot proliferate and contribute to the rare fraction of the community. Above numbers (in red) report the contribution (in percentage) of the dominant lineages to the final community biomass. b) Mean lineage cell age depending on their classification into rare and dominant shows an overall older cell population for the rare fraction due to their lower reproductive success. c) Mean lineage generation time in relation to their relative abundance (log) for three hydration conditions. The rare fraction is typically associated with longer mean generation times and only occurs in drier conditions where aqueous nutrient diffusion is limited to thin water films and capillary pinning forces immobilize bacterial cells.

$\alpha$ and scale parameter $\beta$. The two parameters can be related to the mean generation time of the population by assuming a constant coefficient of variation CV (that is, the standard deviation ($\sigma$) around the mean generation time ($\mu = \tau$) grows proportionally with the generation time $CV = \sigma/\mu$). For the Gamma distribution where $\mu = \alpha\beta$ and $\sigma = \alpha\beta^2$, this results in a constant shape factor $\alpha = 1/CV^2$ and a scale parameter that is linked to the mean generation time $\beta = \tau/\alpha$. We can express the mean generation time in terms of the growth rate $k = \ln(2)/\tau$ to derive an expression for cell age distribution in an isogenic population using Gamma distributed generation times that is only a function of the growth rate k:

$$u(a) = 2 \cdot k \cdot e^{-ka} \cdot S(a, k) \qquad (2)$$

where $S(a, k)$ is the survival function of the Gamma distribution and acts to erode the tail of the exponential distribution as shown for different coefficients of variation in Fig 5A. When

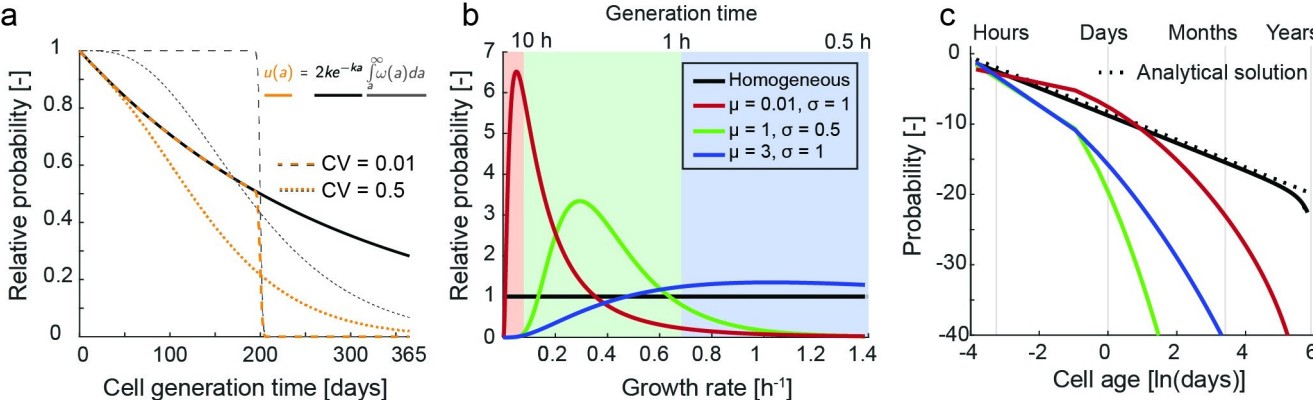

**Fig 5. Numerical integration of the heuristic model to obtain bacterial community cell age distributions.** a) Visualization of the components in Eq (2) for different CV with showing the exponential component (solid black line), the Gamma survival function (dashed gray lines) and resulting cell age distribution for a single population with a mean generation time of 200 days growing in homogeneous conditions (dashed yellow lines). b) Distribution of weights used for the numerical integration of Eq (2) taken from a lognormal distribution to represent different growth regimes including slow growth (red line), intermediate growth (green line) and rapid growth (blue line). The solid black line represents the homogeneous distribution of growth rates. c) Resulting community cell age distribution when numerically integrating Eq (2) using a CV of 0.1 and the weight distribution in panel b. The dashed line represents the power law of Eq (4) assuming a CV of 0 and homogeneous growth rate distribution whereas the solid black line shows the numerical integration when considering a CV of 0.1 and homogeneous growth rate weights.

the CV becomes very small (i.e. where all bacterial cells have exactly the same generation time) the survival function is 1 for $a < \tau$ and 0 for $a > \tau$ and Eq (2) can be modified to:

$$u(a) = \begin{array}{ll} 2 \cdot k \cdot \exp(-k \cdot a) & \text{for } a < \tau \\ 0 & \text{for } a > \tau \end{array} \tag{3}$$

Finally, in contrast to the derivation above for a single population growing in homogeneous conditions, bacterial communities in a soil volume may experience a broad spectrum of growth rates. To obtain the cell age distribution of a soil bacterial community, we integrate Eq (3) with respect to the growth rate k which results in a power law distribution of cell ages for the entire community.

$$\bar{u}(a) = \frac{2}{a^2} \tag{4}$$

The derivation of Eq (4) assumes equal generation times for all cells ($CV \rightarrow 0$) and integrates across all possible growth rates but make the assumption that the growth rates are equally present within the habitat–an assumption that does not reflect conditions within soil. To assess how heterogeneous growth rate distributions may shape the resulting cell age distribution, we numerically integrate Eq (3) using a CV of 0.1 (i.e. the standard deviation equals 10% of the mean generation time) and lognormally distributed weights for the growth rates from a minimum value representing a division once a year to a maximum growth rate that reflects a cell division every 30 minutes (Fig 5B). The three parameterizations were chosen to reflect different potential regimes ranging from predominantly slow growth (red line) via a more balanced growth (green line) to predominantly rapid growth (blue line). We use the homogeneous weights (black line) to compare the numerical integration to the analytical solution (dashed line in Fig 5B assuming a CV of 0). Fig 5C shows the numerically integrated community cell age distributions when taking into account the weights from Fig 5B. Evidently, the weighted growth rates result in a general erosion of the tail of the cell age distribution.

## Discussion

To bridge the information gap due to limitations of current experimental methods in resolving soil bacterial age distributions within individual soil samples, we use a mechanistic model that represents physical microhabitats and simulates individual bacterial cells interacting with their environments to track life histories of cell lineages and estimate generation time distributions. Simulation results show a broad range of growth rates marked by extremely slow growth rates (despite considering a single obligate aerobic bacterial species with an optimal generation time of 0.5 h) where some cells did not divide during the entire duration of the e simulations. These "old" individuals persist by balancing metabolic activity and cell maintenance, a mechanism congruent to experimental observations in batch cultures where cells entered a deep starvation mode in response to diminishing nutrient conditions [35]. Analogous observations of extremely slow growth have been made in fertile soils where average population doubling times exceeding 100 days are common and reproducible [15], with experiments in harsher conditions suggesting that bacterial generation times may become indefinitely long as nutrient interception balances cell maintenance rates [36–38].

The broad and heavy-tailed age distributions observed in the simulations resemble a power law with varying slopes for a range of bacterial cell age with prominent truncated tails of the distributions that vary with hydration conditions (Figs 3 and S1) marking an upper limit for the oldest cells in the simulated domain. The key ingredient for the emergence of these heavy-tailed distributions is the broad range of growth rates within the system under consideration. This assumption is difficult to validate *in situ*, since current methods to determine growth rates in soil measure a sample average growth rate (which would bias the observed growth rate towards rapid growing cells) and cannot resolve the growth rate distribution at the individual cell level. In the model, the wide distribution of growth rates is a consequence of the patchy nutrient landscape dictated by thin water films, especially in dry conditions (S2 Fig) for which the simulation data show a generation time distribution following a power law more closely (Fig 3D). A consequence of such distribution is that no simple mean bacterial generation time can be defined due to the heavy-tailed distribution. We may define the average cell age of certain fractions of the population (S3 Table), such estimates would however be highly biased towards the younger and more abundant cells.

Heavy-tailed generation time distributions promote a division of the bacterial community into dominant and rare lineages. By virtue of their position relative to nutrient sources, dominant lineages that contribute most to the overall biomass (but constitute the minority of the original inoculated cell lineages) proliferate and achieve high reproductive success, which translates to shorter generation times and younger cells on average. In contrast, the rare lineages grow very slowly, thereby exhibiting overall longer generation times and aged cells. Thus, considering a soil volume that contains at least a single hotspot, we expect the emergence of a bimodal distribution of cell ages (Figs 2 and S3) where the interplay of bacterial hot and cold spots as a function of diffusion limitations provide mechanisms that support the emergence of within species diversity [39]. In addition, an interesting analogy exists between the rare and dominant lineages in our simulations and the rare and common species found in natural soil [40,41]. Recent experimental results have provided glimpses into a wide distribution of taxon-specific growth rates for the same soil sample [42], raising the question of whether bacterial cold spots (representing the rare lineages) also harbor the tail of the bacterial species distribution (rare species). This discrepancy in growth dynamics between lineages (or strains/species) culminates in the creation of a "genetic reservoir" (also termed microbial genetic seed banks [43]) where rapid proliferation generates genetic variation that persists through time owing to the slow growing and rare lineages that act as trait keepers. Often such genetic reservoirs have

been associated with the fraction of the bacterial population at a state of reduced metabolic activity (dormancy) within larger soil volumes [44]. Our analysis suggests that even under ubiquitous hydrological conditions and within small soil-volumes (mm$^3$) occupied by a single bacterial hotspot, the persistence of rare lineages existing close to the maintenance rate provides a simple and intuitive mechanism for the emergence of heavy tailed generation time distributions that contribute to the soil bacterial genetic reservoir.

The question remains how important the soil microbial genetic reservoir is for ecosystem stability and function. The dynamic nature of soil as a microbial habitat creates scenarios where adapted gene variants are outcompeted due to changing environmental conditions. We hypothesize that under conditions where environmental conditions revert to a previous state, such genetic memory within every bacterial species may promote rapid adaption of soil microbial communities [45]. Evidence for the importance of the microbial seed bank in soil has been found in the disproportionate response of rare taxa during following rapid changes in environmental cues [46] or during a controlled 45-week experiment including a thermal disturbance [47]. In both cases, resuscitation of dormant taxa from the vast soil genetic reservoir was key for ecosystem stability and function.

Admittedly, the models describe a highly abstracted and simplified reality representing a small domain with limited heterogeneity where a single, aerobic species growing on a sole carbon source in absence of environmental variables (e.g. pH or temperature) with the exception of hydration conditions. Evidently, microbial life in soil is characterized by a vast diversity in traits that govern the generation time distribution. For instance, cell-size variation across species plays a critical role in the metabolic rate and may thus significantly influence the generation time distribution for different cells [48]. Motility is an important trait for soil bacterial species. In comparison to motile obligate aerobic species, non-motile bacterial species produce small colonies that are exposed to local growth conditions and unable to relocate towards more favorable conditions. Individual colonies are limited by nutrient flux through the aqueous phase and cannot self-engineer the overall nutrient landscape of the hotspot, resulting in a homogenization of the generation time distribution across hydration conditions (S4 Fig). In these simulations, only saturated conditions are significantly different from the drier conditions due to the low oxygen diffusion through the saturated pore space. We further investigated how removing the oxygen dependency for growth (essentially simulating motile facultative anerobic species) changes the generation time distribution for all hydration conditions (S5 Fig). In this case, saturated conditions (-0 and -1 kPa) resulted in a vastly higher overall population size and shift in generation time distribution towards shorter generation times when compared to the obligate aerobic species simulations. For unsaturated conditions, the changes in population size and generation time distributions are minimal, confirming that the system is governed by diminishing carbon diffusive fluxes. In contrast to out simulations, bacterial communities in soil hotspots with oxygen and carbon counter-gradients may self-segregate into sub-populations following their oxygen preference [49] and engage in cross-feeding of intermediate metabolites as demonstrated *in silico* [28] and experimentally in bacterial colonies growing on agar plates [50] or microcolonies within microfluidic devices [51]. This scenario would lead to more rapid proliferation of anaerobically growing cells in close proximity of the carbon source whereas obligate aerobic species are restricted to leaking intermediate carbon sources with limited prospects for rapid growth. Based on these results, we would expect a further shift of the generation time distribution in saturated conditions converging towards the distribution observed in intermediate hydration conditions with a greater abundance of rapidly growing cells and a slightly more pronounced tail of the distribution due to aerobic cells growing on intermediate carbon sources. In addition to different bacterial traits, natural soil conditions impose numerous further constraints such as highly complex pore

spaces, restricted nutrient accessibility, a plethora of potential carbon sources and additional biotic factors (such as intraspecific variability and non-growth associated maintenance). Since our simulations represent the optimal case of rapid bacterial growth, we expect that the above-mentioned processes likely extend the generation time distribution and contribute further to the genetic reservoir in soil [44].

In analogy, we also expect a range of biotic and abiotic processes to act in the opposite direction that would truncate the range of the observed heavy-tailed generation time distributions. The lowest limit for the age range is defined by physiological constraints affecting rapid bacterial growth (i.e. the shortest metabolically supported generation time), whereas mechanisms that truncate the upper range of the distributions are more complicated to determine. Unlike mammals, the lifetime of a bacterial cell is not limited [36–38]. However, there are other mechanisms of bacterial death which shape the cell age distribution, such as grazing of bacteria [52], large-scale bacterial death associated with episodic wetting events [53] or phage infections [54] supporting the "forever young" hypothesis [55]. If the mechanisms above act uniformly on the total soil bacterial population, they would affect primarily the dominant lineages (due to their high relative abundance), yet, over extended periods we expect gradual erosion of the age distribution tail and rejuvenation of the overall community. Simulating all potential conditions using the mechanistic model is nigh impossible due to computational limitations. However, the heuristic model with the assumption of heterogeneous growth rate distributions can be used to estimate the influence of how a shift in hydration conditions may reshape the expected generation time distribution. If for instance a rainfall event shifts the local growth rate distribution towards faster growth rates (e.g. from the red line to the green or blue in Fig 5B), the expected cell age distribution would become skewed towards younger cells for a brief period of time until water drains and soil field capacity is reached (often within a few hours) which we verified using a numerical simulation containing dynamic moisture conditions (S6 Fig). The bacterial generation time distributions are thus expected to revert to their characteristic shape for the soil under consideration. Importantly, although such events result in an overall rejuvenation of the extant community, they do not diminish the genetic generational gaps that have emerged during the previous growth phase. However, we emphasize that results of the heuristic model need to be interpreted in light of the assumptions of the underlying equations outlined in the results section. Overall, since many of the above-mentioned mechanisms are related to the microscale liquid organization in soil pores, the resulting bacterial cell age and related generation time distributions represent a delicate balance between processes that promote and suppress survival of old bacterial cells as a function of soil hydration conditions.

To resolve the bacterial age and generation time distribution of small soil volumes, novel experimental approaches are required that manage to capture the breadth of bacterial growth rates. A promising avenue is to use chromosomal barcoding of bacterial species that enables direct tracking of individual lineages [56,57]. In this approach, a single bacterial species is tagged with a library of unique genetic barcodes that enable to track individual lineages throughout time and observe the emerging generation time distribution. Furthermore, by combining this approach with whole genome sequencing, the influence of individual mutations on the population composition and overall evolution of the initially isogenic population can be resolved [58,59] and sheds light onto the missing link between individual bacterial population dynamics and their generation time distributions with the overall population evolution.

The original aim of estimating bacterial cell age and generation time distributions in soils in relation to heterogeneity and hydration conditions began with a very simple question: "What is the average age of a soil bacterial cell?". In contrast to macrobiota with a finite life

span where an average age in an ecosystem is well defined, providing a similar answer for pro-karyotes living within heterogeneous soil is far more complicated. Overall, our results suggest that the emerging bacterial generation time distribution in soil is a function of the soil hydration state and related patchy nutrient conditions, and the tails of the generation time distribution are likely to be eroded by various biotic (bacteriophages, grazing by protists) and abiotic (osmotic bursting upon rewetting, starvation due to diffusion limitations) soil and climatic processes. The proximity of vastly different ages and generations of the same bacterial species that may coexist a few hundred microns apart raise several intriguing possibilities. Considering reconnection of subpopulations with large generational gaps by episodic soil wetting events, could offer opportunities for regaining physiological traits lost during prolonged segregation and potentially provide a ubiquitous mechanism for sustaining genetic reservoir of traits and ecotypes [39].

## Methods

### Mechanistic modeling of bacterial age and cell lineages in soil

The spatially explicit mechanistic modeling framework (IndiMeSH) was used to simulate bacterial life in heterogeneous soil microhabitats [28] mimicking a soil aggregate (or a hotspot around an active root segment, Fig 2). The domain is comprised of an angular pore network occupying a spherical soil cross section (10 mm radius) containing individual pore channel segments with lengths of 100 microns (106,901 pores). The pore channels with triangular cross sections are drawn from a uniform distribution of central angles between 30° and 150° with inscribed pore radii sampled from a lognormal distribution with mean 30 microns and variance 10 microns. In addition, systematic variations of the domain heterogeneity were included by varying pore sizes in 50 randomly located regions of the pore network. Hydration conditions are prescribed by a matric potential that translates into a distribution of micro-aqueous habitats based on the pore network (see Borer et al. 2019 [28] for detailed equations). The effective water film thickness is derived for each individual pore channel based on its geometry and the prescribed matric potential. Bacterial cells are simulated as individual agents that are motile following a run-and-tumble mechanism including a chemotactic bias. A constant carbon source is located at the center of the simulation domain (0.1 mM) with oxygen sources at the periphery of the pore network, following Henry's law (constant source of 0.27 mM). This arrangement of boundary conditions gave rise to counter gradients of oxygen and carbon mimicking conditions frequently found in natural soil hotspots [3]. For simplicity, we modeled bacterial cells as obligate aerobes, with Monod kinetics including carbon and oxygen limitation terms. Under optimal conditions (no oxygen or carbon limitation), the simulated bacterial cells have a mean generation time of 28 minutes. We focus on aerobic growth since this is the most common state in most near surface soils. In some soils and under certain conditions, saturated conditions throughout the soil profile may prevail, however, for simplicity and considering the long-time horizons of the analyses, we neglected these cases. All model parameters concerning bacterial growth are shown in S2 Table and are based on Borer et al. 2018 [49]. A total of 1000 bacterial cells are inoculated homogeneously across the domain (representing an isogenic population where each cell has the same growth parameterization) with a total simulation time of 30 days at 10 s time steps. The age of each individual bacterial cell is captured as the time since last division. For each dividing cell, its current age is stored as the generation time while resetting the age of the daughter cells. A unique identifier (similar to a genetic barcode) is assigned to each inoculated cell that is inherited by its progeny, enabling tracking of reproductive success, generation time distribution and cell age distribution of each lineage. Each hydration condition was simulated in triplicates. However, inter-replicate variation was

negligible (despite containing stochastic elements, the large number of simulated individual cells homogenize the results across replicates).

## Supporting information

**S1 Fig. Long tailed distributions fitted to the IndiMeSH simulation data using maximum likelihood estimation.**
(PDF)

**S2 Fig. Cell dispersal and diffusion regime determine bacterial generation time distribution.**
(PDF)

**S3 Fig. Mean growth rates of individual lineages through time for two contrasting hydration conditions.**
(PDF)

**S4 Fig. Influence of cell motility on bacterial proliferation and generation time distributions.**
(PDF)

**S5 Fig. Comparison of simulation between obligate aerobic and oxygen independent (facultative anaerobic) species.**
(PDF)

**S6 Fig. Dynamic hydration conditions and resulting generation time distribution.**
(PDF)

**S1 Table. Experimental details of data used in Fig 1.**
(PDF)

**S2 Table. Parameters used in the mathematical model (IndiMeSH).**
(PDF)

**S3 Table. Simulated mean cell age [days] as a function of matric potential for different cumulative biomass cutoff threshold.**
(PDF)

**S1 Data. Initialization files and IndiMeSH model.**
(ZIP)

## Acknowledgments

We thank Noah Fierer (University of Colorado, Boulder) for motivating this work with the question "what is the average age of soil bacteria?" We thank Robin Tecon and Samuel Bickel (ETH, Zurich) for discussing the presented material on numerous occasions.

## Author Contributions

**Conceptualization:** Benedict Borer, Dani Or.

**Data curation:** Benedict Borer.

**Formal analysis:** Benedict Borer.

**Funding acquisition:** Dani Or.

**Investigation:** Benedict Borer.

**Supervision:** Dani Or.

**Visualization:** Benedict Borer, Dani Or.

**Writing – original draft:** Benedict Borer, Dani Or.

**Writing – review & editing:** Benedict Borer, Dani Or.

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
