## [Decision Letter · Decision Letter 0]

21 Jun 2021

Dear Dr Borer,

Thank you very much for submitting your manuscript "Bacterial age distribution in soil – generational gaps in adjacent hot and cold spots" for consideration at PLOS Computational Biology.

As with all papers reviewed by the journal, your manuscript was reviewed by members of the editorial board and by several independent reviewers. In light of the reviews (below this email), we would like to invite the resubmission of a significantly-revised version that takes into account the reviewers' comments.

We cannot make any decision about publication until we have seen the revised manuscript and your response to the reviewers' comments. Your revised manuscript is also likely to be sent to reviewers for further evaluation.

Sincerely,

Jacopo Grilli

Associate Editor

PLOS Computational Biology

Alice McHardy

Deputy Editor

PLOS Computational Biology

Reviewer's Responses to Questions

**Comments to the Authors:**

Reviewer #1: Overview:

This paper explores how growth rates of bacteria are affected by soil moisture using individual based modeling (IBM). The model, which has previously been described in published papers, simulates physical aspects of the soil environment at scales that are relevant to the physiology of cells in terms of motility, access to substrates, and oxygen availability. Simulation across a range of matric potentials reveals that total abundance is maximized at intermediate moisture contents. The paper then examines the distribution of generation times with respect to water content and reports that they follow a power-law distribution such that a few younger lineages (in “hot spots”) contribute most to total abundance, yet many lineages (in “cold spots”) are collectively rare, accounting for less than 3% of the population. The model is based on principles and reasonable assumptions that will make sense to people who think about microorganisms living in the soil environment. What I like most about the paper is that it makes some conceptual extensions about the life history, demographics, and longevity of populations that are not often considered by soil ecologists.

It seems that a number of other papers have been published using the IBM platform presented in this paper. On the one hand, this means that the approach itself is not especially novel, nor does it represent a particularly major methodological advancement. On the other hand, it means that the model has been vetted to some degree. Therefore, this evaluation tends to focus more on the application of the IBMs to the major finding, which is that the life span of bacterial cells is predicted to follow a power law distribution. The implications are that in a typical gram of soil, young and old cells may be found together.

Major comments:

--In Figure 1, the literature reported data has a longer tail, and the simulations seem to have a Poisson distribution. If growth reflects births, then one might expect a Poisson process giving rise to times between generation that follow an exponential distribution, which would have implications for the inferences that are being made. It’s not clear if other distributions were considered, but a more rigorous treatment along these lines is recommended.

--Evolution is invoked in various parts of the paper. For example, the abstract talks about “evolutionary milestones” and the “differential pace of evolution”. Unless I missed some critical details, there are no evolutionary processes in the IBMs, unless one wants to view motility as gene flow, but this is not explicitly developed. There is not genetic or phenotypic variation among individuals, that might be reflected by mutation, selection, or drift. While I think there are interesting implications for variation in growth rate for the evolution of populations, the model seems to be inadequate for making such statements. I’m concerned that a casual reader might be led to think and cite the paper based on claims that are not rigorously assessed with the model. Thus, I think that the evolutionary claims either need to be removed, or the manuscript and model need to be presented in a more thorough way.

--The paper attempts to make fairly wide-reaching conclusion about the lifespan of cells, but only in soils. Are the findings only relevant to soils? If so, why? Some of the features that seem to be implicated in longevity relate to spatial heterogeneity, but this is characteristic of many microbial habitats. What about other taxa? It would be useful if the paper would more explicitly consider the generality (or lack of) of the models and inferences that are made from them.

--It is well known that water content in soils fluctuates through time. While the frequency may differ according to geographic region (e.g., desert vs. temperate grassland), it’s reasonable to assume that water content varies through time, which may “erase”, disrupt, or reset the power law relationship when cold sites become hot sites, thus preventing the accumulation of old-aged individuals. Such features seem to be already be incorporated into the model, but instead, the paper seems to focus on static conditions held at different water potential. I think discussion and perhaps additional simulations might be warranted here.

--The results section of the paper attempts to provide mechanistic explanations for the power law distribution of generation times by invoking the contributions of specific physical forces and physiological limits. Perhaps it makes sense not to monkey around with soil physical properties, but there’s a lot of variation in traits of microbes in real soils. For example, many bacteria in soil are facultatively anaerobic, and many populations exhibit variation in motility. If strong claims are to be made about how oxygen and motility lead to the emergence of power law distributions in lifespan, wouldn’t it be reasonable to test this with simulations where these traits are directly manipulated/controlled?

--In the discussion, the paper talks about “unbounded range of bacterial ages”. The paper asks what the age of a bacterial cell might be. The answer provided is “practically any age”, suggesting there is no limit. Perhaps this comes from a strict interpretation of the power law based on simulations corresponding to on 30 days. But there are good reasons to believe that there are real constraints on cell age, particularly in soils, where microbes commonly experience stress (oxidative, nutritional, radiation, grazing) in their characteristically dynamic environment.

Minor comments:

--Overall, I think the paper would benefit from some polishing of the text. The writing was sometimes hard to follow. Some examples include incomplete sentences (e.g., lines 29-31). In other places, the paper uses terminology that was confusing. For example, “coexisting” (line 37) conjures up processes and theory relating to diversity maintenance, which I don’t see as being relevant given my understanding of the model structure. Similarly, there is discussion about the “emergence of relic traits”, which again seems to be outside the scope of what was represented in the IBMs. It would be advisable for any future versions of the paper to be more careful with such language.

--The Figure 2 caption could be improved for clarity. The methods section does a good job of explaining, but I could not understand in a “stand alone” way what was going on in panel b.

--In Figure 3 panel c, it does not seem like the power law models does a good job of fitting the simulated data for wet soil. Given this, I think it would be appropriate to conduct a more formal analysis. In other words, does a power law do a good job of describing the distribution of generation times for all moisture conditions? Should other models be considered in some cases? This is addressed to some degree in the supplement, but is less obvious in the discussion and figures in the main text.

--Could the authors expand their discussion with regard to how the model predictions could be explicitly tested in an experimental system, or better yet in the real world? Some approaches are alluded to in the introduction, but it seems like it would be appropriate to help guide readers towards ways of evaluating.

Reviewer #2: Please se attached

**Have the authors made all data and (if applicable) computational code underlying the findings in their manuscript fully available?**

Reviewer #1: Yes

Reviewer #2: Yes

PLOS authors have the option to publish the peer review history of their article (what does this mean?). If published, this will include your full peer review and any attached files.

Reviewer #1: No

Reviewer #2: **Yes: **Samraat Pawar & Jacob Cook
---

## [Decision Letter · Decision Letter 1]

11 Oct 2021

Dear Dr Borer,

Thank you very much for submitting your manuscript "Bacterial age distribution in soil – generational gaps in adjacent hot and cold spots" for consideration at PLOS Computational Biology.

As with all papers reviewed by the journal, your manuscript was reviewed by members of the editorial board and by several independent reviewers. In light of the reviews (below this email), we would like to invite the resubmission of a significantly-revised version that takes into account the reviewers' comments.

We cannot make any decision about publication until we have seen the revised manuscript and your response to the reviewers' comments. Your revised manuscript is also likely to be sent to reviewers for further evaluation.

Sincerely,

Jacopo Grilli

Associate Editor

PLOS Computational Biology

Alice McHardy

Deputy Editor

PLOS Computational Biology

Reviewer's Responses to Questions

**Comments to the Authors:**

Reviewer #1: Overview: This is an evaluation of revised manuscript that I made comments on previously. In general, it seems like the authors have made attempts to correct some of the major issues that were identified, but have elected not to explore some of the other topics that were raised by both reviewers. So, perhaps I’m left a little bit less excited about the paper, but as stated previously, I still think the paper is valuable in that the simulations generate some interesting patterns that could perhaps be experimentally tested.

As noted before by both reviewers, the abstract makes some bold claims. The authors state that they have toned things down, but there still seems to be some disconnect. Also, one of my major concerns about the paper was that it was inappropriately making reference to evolution. The authors indicated that they eliminated this focus, but then, in the abstract, the paper is talking about the role of beneficial mutations. Why? This is not within the scope of the paper based on what the model does. While it may be reasonable to talk about how future versions of the model could explore the evolution of lineages, it is not appropriate in the current model which is made up of isogenic, non-mutable individuals in the absence of selection. At the end of the abstract, there’s reference to the emergence of a genetic reservoir. Again, there’s no genetic variation among the individuals. I’m a bit puzzled and feeling misled.

It seems that the authors have side-stepped an important issue raised by both reviewers. Soil moisture is highly variable through time and has implications for the primary pattern of the paper: that is, a heavy-tail age distribution of cells within a smaller area. I asked whether the frequent rewetting of soil would erode this pattern. The other reviewer asked about the time-scale of mixing owing to periodic precipitation (i.e., hot and cold spots change over time). The authors argue that this is not relevant because of the sufficient discrepancy in time scales between rain events and growth rates, but this doesn’t make sense to me. The authors argue that there are 140 rain days in Zurich. If rainfall is more or less equally distributed across the simulated area of habitat, this would mean that cells in soils are experiencing rewetting (and growth) every 2.6 days. This falls right within the distribution of (measured and simulated) generation times of soil bacteria (Fig. 1). Yet, simulations seem to run for ~30 days without precipitation (Fig. 2). Given this, I think it is very reasonable to ask how temporal fluctuations in soil moisture affect the heavy-tail age distribution. Are rain events in mesic ecosystems (like those in Zurich), homogenizing growth rates on times scales that make the reported heavy-tail age-distribution transient, or perhaps even trivial? Or, maybe these patterns are more important in arid ecosystems. In any case, it seems like it wouldn’t be hard to run such a simulation. I inquired about the generality of the pattern, and the authors argued that the pattern was most relevant to soils. That is fine, but if the work is meant to capture important features of soil ecosystems, then maybe it would be good to address one of its most universal aspects: soil moisture is temporally dynamic.

The authors also did not seem interested in exploring variation in functional traits that underly important aspects of demography in environments with different moisture levels, namely motility and tolerance to oxygen. The authors responded that exploring variation in these (and others) is “nigh impossible”. But it seems like the models used here are the perfect tool for exploring this type of question. For example, what would the heavy-tail age distributions look like if bacteria were not obligately aerobic? Perhaps I am underestimating the amount of work it would take to address, and I’m suggesting that this needs to be done. But by electing to explore these processes, there ends up being a disconnect between the bold generalizations and what was actually done with the models.

Reviewer #2: The authors have made extensive revisions to address the points rauised by both reviewers. Overall, we are satisfied with the revisions, except one point: We appreciate the fact that the relative tractability of a single species (but multi-lineage) model provides mechanistic insights that a more complex community model wouldn't. However, it is hard to argue that a single species can produce a generation time distribution that is comparable to that generated by multiple species in a community. This is partly because cell-size variation is greater across rather than within species in microbes (as is typically the case in other organismal groups). Now it may well be that for community of multiple lineages of a single species, spatial/environmental structure will produce a generation time distribution that is *qualitatively* the same as that generated by a multi-species community. But this cannot be determined from the current theoretical framework as such. Therefore, we encourage the authors to address this disparity (in words) in their paper's discussion.

**Have the authors made all data and (if applicable) computational code underlying the findings in their manuscript fully available?**

Reviewer #1: **No: **this should be checked; authors statement on code availability just directed to a url for a paper that was previously published in PLOS Computational Biology

Reviewer #2: Yes

PLOS authors have the option to publish the peer review history of their article (what does this mean?). If published, this will include your full peer review and any attached files.

Reviewer #1: No

Reviewer #2: **Yes: **Samraat Pawar
---

## [Decision Letter · Decision Letter 2]

23 Jan 2022

Dear Dr Borer,

We are pleased to inform you that your manuscript 'Bacterial age distribution in soil – generational gaps in adjacent hot and cold spots' has been provisionally accepted for publication in PLOS Computational Biology.

Best regards,

Jacopo Grilli

Associate Editor

PLOS Computational Biology

Alice McHardy

Deputy Editor

PLOS Computational Biology

Reviewer's Responses to Questions

**Comments to the Authors:**

Reviewer #1: I appreciate the authors' time and and effort to address concerns raised in the last set of views. I think this has led to a more balanced and improved study with some interesting patterns for follow-up work.

Reviewer #2: I think the authors have done a good job generating further results, and rewording the mansucript to make it more faithful to the scope of the results. The paper will stimulate further work in this very interesting and potentially important line of investigation. I am happy to recommend acceptance.

**Have the authors made all data and (if applicable) computational code underlying the findings in their manuscript fully available?**

Reviewer #1: Yes

Reviewer #2: **No: **It is stated in the manuscript that the code used for generating all data displayed in this manuscript is available online (https://doi.org/10.1371/journal.pcbi.1007127). However, this inks to their 2019 paper. For the sake of reproducibility the code used to generate the results specific to this paper needs to be shared.

PLOS authors have the option to publish the peer review history of their article (what does this mean?). If published, this will include your full peer review and any attached files.

Reviewer #1: No

Reviewer #2: **Yes: **Samraat Pawar & Jacob Cook

---

## [Editor Report · Acceptance letter]

22 Feb 2022

PCOMPBIOL-D-21-00669R2 

Bacterial age distribution in soil – generational gaps in adjacent hot and cold spots

Dear Dr Borer,

I am pleased to inform you that your manuscript has been formally accepted for publication in PLOS Computational Biology. Your manuscript is now with our production department and you will be notified of the publication date in due course.

With kind regards,

Katalin Szabo
